# Comparison of Microbiological Characteristics and Genetic Diversity between *Burkholderia cepacia* Complex Isolates from Vascular Access and Other Clinical Infections

**DOI:** 10.3390/microorganisms9010051

**Published:** 2020-12-27

**Authors:** Min Yi Wong, Yuan-Hsi Tseng, Tsung-Yu Huang, Bor-Shyh Lin, Chun-Wu Tung, Chishih Chu, Yao-Kuang Huang

**Affiliations:** 1Division of Thoracic and Cardiovascular Surgery, Chiayi Chang Gung Memorial Hospital, Chiayi County 61363, Taiwan; mynyy001@gmail.com (M.Y.W.); 8802003@cgmh.org.tw (Y.-H.T.); 2Microbiology Research and Treatment Center, Chiayi Chang Gung Memorial Hospital, Chiayi County 61363, Taiwan; p122273@cgmh.org.tw; 3Institute of Imaging and Biomedical Photonics, National Chiao Tung University, Tainan 71150, Taiwan; borshyhlin@gmail.com; 4College of Medicine, Chang Gung University, Taoyuan City 33302, Taiwan; Wood5@ms55.hinet.net; 5Division of Infectious Diseases, Department of Internal Medicine, Chiayi Chang Gung Memorial Hospital, Chiayi County 61363, Taiwan; 6Department of Nephrology, Chiayi Chang Gung Memorial Hospital, Chiayi County 61363, Taiwan; 7Department of Microbiology, Immunology and Biopharmaceuticals, National Chiayi University, Chiayi City 60004, Taiwan; cschu@mail.ncyu.edu.tw

**Keywords:** *Burkholderia cepacia* complex (BCC), *B. contaminans*, *B. cenocepacia*, non-CF, *recA* sequencing, multilocus sequence typing (MLST)

## Abstract

*Burkholderia cepacia* complex (BCC) is a group of closely related bacteria with widespread environmental distribution. BCC bacteria are opportunistic pathogens that cause nosocomial infections in patients, especially cystic fibrosis (CF). Multilocus sequence typing (MLST) is used nowadays to differentiate species within the BCC complex. This study collected 41 BCC isolates from vascular access infections (VAIs) and other clinical infections between 2014 and 2020. We preliminarily identified bacterial isolates using standard biochemical procedures and further conducted *recA* gene sequencing and MLST for species identification. We determined genetic diversity indices using bioinformatics software. We studied 14 isolates retrieved from patients with VAIs and observed that *Burkholderia cepacia* was the predominant bacterial species, and *B. contaminans* followed by *B. cenocepacia* were mainly retrieved from patients with other infections. According to MLST data, we identified that all *B. contaminans* isolates belonged to ST102, while a wide variety of sequence types (STs) were found in *B. cenocepacia* isolates. In summary, the high diversity and easy transmission of BCC increase BCC infections, which provides insights into their potential clinical effects in non-CF infections.

## 1. Introduction

*Burkholderia cepacia* (formerly *Pseudomonas cepacia*) was previously known as one bacteria species, but it has expanded to the *Burkholderia cepacia* complex (BCC), which comprises more than 20 closely related opportunistic pathogenic species [1,2]. BCC is a group of genetically distinct but phenotypically similar bacteria that occur naturally in remarkably diverse ecological niches and possess extraordinary metabolic versatility [3,4,5]. Despite their agricultural potential, BCC is also recognized among opportunistic nosocomial pathogens, with high potential to cause infections in patients with compromised immunity, cystic fibrosis (CF), chronic granulomatous disease, and indwelling medical devices [6,7]. Among the complex species, *B. cenocepacia* and *B. multivorans* are the predominant pathogens that account for about 85–97% of all BCC infections in CF [8], causing a severe decline in lung function, possibly further developing into cepacia syndrome. Nevertheless, there is growing clinical interest in *B. contaminans,* which was previously identified as taxon K, and this interest has been associated with CF patients, especially in Argentina and Spain [9,10]. *B. contaminans* was first identified from a contaminated Sargasso Sea DNA sample and was designated as *Burkholderia* SAR-1 metagenome at that time. The species is epidemiologically linked to the contamination of medical devices and pharmaceutical products, such as nasal sprays, dialysis water, intravenous fentanyl, and moist washcloths [11,12,13,14], causing healthcare-associated infections.

Since conventional phenotypic identification of the closely related BCC species is often difficult and not straightforward, 16S rRNA gene sequence analysis was introduced as a useful tool to discriminate BCC bacteria [15,16]. However, as 16S rRNA gene analysis has limited taxonomic resolution that exclusively discriminates BCC at the complex level, 16S rRNA has been surpassed by *recA* gene sequence analysis as a means to identify BCC at species level [17]; however, *recA* gene sequence analysis offers limited resolution. Improved resolution in strain identification is offered by multilocus sequence typing (MLST), the gold standard for typing both species and strains within the species, and this provides new insights into the population structure. The ability of MLST to differentiate existing BCC species is greater compared to analysis of the *recA* gene alone, and it is a valuable approach in offering a high level of strain identification without using polyphasic taxonomy techniques [18]. 

Here, we conducted a six-year single-institution study to (a) investigate the prevalence of BCC species from vascular access infections and other BCC infections (non-CF), (b) determine the genetic diversity and multilocus sequence of different bacterial species isolated from different infections and specimens, and (c) establish correlation between BCC species, sequence types (STs), and infection types.

## 2. Materials and Methods

### 2.1. Ethical Approval

This study was approved by the Institutional Review Board (IRB) of Chang Gung Memorial Hospital (IRB Nos: 201204188B0, 13/05/2013; 2015080482B0, 25/12/2015; and 201801001B1,27/06/2018). Written consent was obtained from patients, and the study was carried out as per approved guidelines.

### 2.2. Study Setting, Bacterial Isolate Collection, and Identification

This single-institution study was conducted at Chiayi Chang Gung Memorial Hospital, a territory referral hospital in Taiwan, from January 2014 to March 2020. We prospectively collected information on ten patients with BCC vascular access infections (VAIs) requiring tunneled cuffed catheter (TCC) removal. Patients with poor compliance or who refused to join the study were excluded. Informed consent was obtained from all patients with a personal explanation before performing the procedures. The other 26 patients with BCC infections from other diseases, including lung disease, reproductive disease, eye disease, kidney disease, and soft tissue infections between July 2018 and March 2020 were also analyzed for comparison. Bacterial isolates were cultured under laboratory standards. The samples were routinely cultured on blood agar at 37 °C overnight. Strain identification was performed using standard biochemical (phenotypic) procedures. The isolates collected after 2019 were identified using matrix-assisted laser desorption/ionization time-of-flight (MALDI-TOF).

### 2.3. Genomic DNA Extraction

A single colony from a clinical isolate was inoculated in tryptic soy broth (TSB) for 16 h, and 1 mL of the overnight culture was harvested by centrifugation at 16,500× *g* for 5 min. Bacterial cells were suspended in 1 mL of ultrapure water and heated at 100 °C for 15 min. The supernatant containing the DNA was stored at 4 °C until further use.

### 2.4. Molecular Characterization

#### *Burkholderia cepacia* Complex (BCC) Species Identification

We identified and determined the species of BCC isolates by *recA* gene amplification and sequencing. We performed PCR amplification using specific primers and conditions described by Fehlberg et al. [19]. Cycle sequencing was performed using the BigDye Terminator v3.1 cycle sequencing kit (Applied Biosystems, Thermo Fisher Scientific Company, Waltham, MA, USA) and an ABI 3730xl DNA analyzer (Applied Biosystems, Thermo Fisher Scientific Company, Waltham, MA, USA). The *recA* sequences were further analyzed and aligned to the NCBI nucleotide Basic Local Alignment Search Tool (BLASTn) database. 

### 2.5. Multilocus Sequence Typing (MLST) Locus Amplification and Sequencing

Among BCC isolates, we conducted MLST by amplifying seven housekeeping genes (*atpD*: ATP synthase beta chain; *gltB*: glutamate synthase large subunit; *gyrB*: DNA gyrase subunit B; *recA*: recombinase A; *lepA*: GTP binding protein; *phaC*: acetoacetyl-CoA reductase; and *trpB*: tryptophan synthase subunit B) using primer sets and PCR amplification conditions as described previously [20]. We designed an alternative forward primer for the undetected *trpB* gene of BCC: trpE-F2, 5′-AAGGACGCGCTGAACGAAGC-3′. We also used alternative primer sets for undetected *atpD*, *gyrB*, *lepA*, and *recA* genes, as a previous study described [21]. Briefly, PCR products were sequenced using the BigDye Terminator v3.1 cycle sequencing kit and analyzed with an ABI 3730xl DNA analyzer. We further aligned the sequences from both strands of a given locus of the same isolate, trimmed to the desired length, and edited using BioNumerics software ver. 7.6 (Applied Maths, Sint-Martens-Latem, Belgium).

### 2.6. Genetic Diversity Indices

The G+C content, number of polymorphic sites, nucleotide diversity (π) (representing the average number of nucleotide differences per site between random pairwise nucleotides), total number of mutations (Eta), number of haplotypes (h), and haplotype diversity (Hd) (defined as the probability that two randomly chosen haplotypes would be different) were calculated using DnaSP Version 6.12.03 [22]. The average nonsynonymous/synonymous substitution rate ratios (*d_N_*/*d_S_*) were calculated using Molecular Evolutionary Genetics Analysis software, MEGA X (Nei and Gojobori’s method) for inferring the direction and magnitude of natural selection [23].

### 2.7. Multilocus Sequence Typing (MLST) and Analysis

We assigned the allele numbers, sequence types (STs), and clonal complexes (CCs) of each isolate using BioNumerics software ver. 7.6. We submitted the novel alleles and STs to the PubMLST database (https://pubmlst.org/) for new allele numbers and ST assignment. We also constructed the minimal spanning tree (MST) with MLST data from the seven housekeeping genes using BioNumerics software ver. 7.6 with the categorical coefficient. 

## 3. Results

In total, we collected 41 *Burkholderia cepacia* complex (BCC) clinical isolates from 36 patients with both VAIs and infections other than VAIs during the six years of the study, starting in January 2014. In five of the patients with VAIs, two BCC isolates from two different types of specimens were collected (i.e., contaminated Hickman catheter tip and blood). The ratio of patients with VAIs to other infections was 1:2.6, with more females than males with VAI and contrary to other infections (Table 1). 

### 3.1. Identification and Prevalence of Burkholderia cepacia Complex (BCC)

The majority of *Burkholderia* species involved in clinical infection in our institution were *B. contaminans*, followed by *B. cepacia* and *B. cenocepacia* (Figure 1). Moreover, based on *recA* gene alignment to the NCBI database, we classified three isolates as “other BCC” due to the high similarity with *Burkholderia* spp. The most common species in VAIs was *B. cepacia*, which was not isolated from other BCC infections (Table 2). *B. cepacia* isolates were derived from contaminated Hickman catheter tip (*n* = 8) and blood (*n* = 3). In contrast, *B. contaminans* was the predominant species in other BCC infections, followed by *B. cenocepacia*. *B. contaminans* isolates were derived from various specimens, with those from the blood (*n* = 4) and endocervix discharge (*n* = 4) being predominant, followed by corneal ulcer (*n* = 2), percutaneous nephrostomy (*n* = 2), pus (*n* = 2), Hickman catheter tip (*n* = 2), and central venous pressure tip (*n* = 1). *B. cenocepacia* isolates were mainly derived from respiratory system specimens (4/9), including sputum (*n* = 3) and bronchoalveolar lavage (*n* = 1). In this study, 12 cases of multiple-species infections (i.e., isolation of BCC, especially *B. contaminans* and *B. cenocapacia*, and other species of bacteria in one specimen) were predominantly found in other BCC infections (11/27), especially from the endocervix discharge of *B. contaminans* infection.

### 3.2. Genetic Diversity Analysis

Table 3 summarizes data on the genetic diversity of the seven loci, including G+C content, number of alleles, number of polymorphic sites, nucleotide diversity (π), total number of mutations (Eta), number of haplotypes (h), haplotype diversity (Hd), and the ratio of nonsynonymous (*d_N_*) to synonymous (*d_S_*) substitutions. The number of alleles observed for each DNA fragment ranged from 7 (*atpD*, *recA*) to 14 (*phaC*), and the most frequently detected alleles at each locus were *atpD* 64 (*n* = 17), *gltB* 80 (*n* = 17), *gyrB* 76 (*n* = 17), *recA* 89 (*n* = 17), *lepA* 105 (*n* = 17), *phaC* 97 (*n* = 17), and *trpB* 70 (*n* = 17), respectively. The number of different alleles present per locus in *B. cenocepacia* was highly diverse, including 3 (*atpD*, *recA*, *lepA*), 4 (*gltB*, *phaC*), 9 (*gyrB*), and 5 (*trpB*). On the contrary, only one allele was present per locus in *B. contaminans*. The frequency of each allele within each bacterial species is shown in Figure 2. 

The average G+C content of the MLST loci ranged from 61.1% for *phaC* to 69.4% for *trpB*. The number of polymorphic nucleotide sites (S) varied from 23 (*atpD*) to 70 (*gyrB*), in which the *gyrB* allele possessed the most polymorphic sites at the nucleotide level. The nucleotide diversity indices (π) of the seven loci ranged from 0.01227 (*atpD*) to 0.04532 (*recA*). The haplotype diversity (Hd) ranged from 0.748 for *atpD* to 0.8 for *gyrB*.

The nonsynonymous to synonymous (*d_N_/d_S_*) ratio measures the level and mode of natural selection acting on protein-coding genes. A *d_N_/d_S_* ratio of <1 indicates negative selection (purifying selection), in which nonsynonymous sites evolved more slowly than synonymous sites during the evolutionary process; a *d_N_/d_S_* >1 indicates positive selection (adaptive selection); and neutral selection is indicated if values are close to 1. The *d_N_/d_S_* ratios for the seven loci varied from 0.01984 (*recA*) to 9.8871 *(gyrB*). The *d_N_/d_S_* ratios were extremely high (*d_N_/d_S_* > 1) in four genes (*atpD*, *gltB*, *gyrB*, and *trpB*), indicating positive selection in the four housekeeping proteins. The *d_N_/d_S_* of *recA*, *phaC*, and *lepA* genes was <1, indicating a purifying selection during the evolutionary process.

### 3.3. MLST and Population Structure

Of the 41 BCC isolates, we identified 14 STs, including seven new ST types (1725, 1726,1863–1866, 1868) (Table 4). The predominant *B. contaminans* species in different specimens of VAIs and various clinical infections all belonged to ST102. Intriguingly, all isolates collected from 22 May to 30 July 2019, were ST102 *B. contaminans*, even those from different types of infections, suggesting that detergent contamination happened during that period. In contrast, the nine *B. cenocepacia* isolates from different specimens of various clinical infections showed high diversity with nine STs, including six unique STs (1725, 1726, 1854, 1865, 1866, and 1868) and four novel alleles (*gyrB* 1202–1203; *trpB* 762; *phaC* 606). The *B. cepacia* isolates belonging to ST1723 and ST1724 were only collected from the blood and contaminated catheter tips of patients with VAIs. The three other BCC isolates belonged to the new ST type (ST1863) with four novel alleles (*atpD* 643; *gyrB* 1204; *recA* 703; *lepA* 788). The MLST minimum spanning tree (MST) of all strains, considering their species and infection cases, is depicted in Figure 3a,b. 

## 4. Discussion

Bacteria comprising the BCC are ubiquitous in the environment and capable of infecting people with CF. However, its pathogenicity is not limited to CF patients. CF, a common autosomal recessive hereditary disease in the Caucasian population, is uncommon in Asia, with only a few cases published in Taiwan [24,25,26,27]. Our six-year study collected 41 BCC isolates from non-CF patients that were from VAIs among hemodialysis patients and other clinical infections. We noted that VAIs were more common than CF in Taiwan. In previous studies, *B. cepacia* was the most prevalent species among UK non-CF patients [28], and *B. cenocepacia* and *B. cepacia* were the leading bacteremia species in non-CF patients in Taiwan [29]. In this study, the predominant species *B. contaminans* was collected from different BCC infections, including VAIs and other clinical infections, followed by *B. cepacia* and *B. cenocepacia,* which were only collected from VAIs and other clinical infections, respectively. *B. contaminans* and *B. cenocepacia* were found in multiple-species infections, especially in endocervix discharge with *B. contaminans* infections. 

MLST is the method of choice for BCC typing due to its ability to differentiate the BCC complex species. In our study, we initially misidentified *B. contaminans* as *B. cepacia* and further differentiated them by using MLST. The major sequence type of *B. contaminans* isolates from VAIs and various clinical infections in our institution belonged to ST102. ST102 may have a global distribution among patients with CF and non-CF; indeed, they are also present in other countries of the European Union, the United States, and Russia, according to data from the PubMLST database (https://pubmlst.org/), which provides a platform to study the epidemiology and biodiversity of BCC bacteria. In contrast to *B. contaminans*, *B. cenocepacia* isolates from different infection specimens showed high genetic diversity, in which six of the novel types were revealed in our single institution, suggesting a high capacity of *B. cenocepacia* to rapidly mutate and adapt to enhance its presence in the patients’ environments or hospital settings. 

In a previous study, Kaitwatcharachai et al. [30] reported catheter-related *Burkholderia cepacia* complex bacteremia in patients undergoing hemodialysis in Thailand, and contaminated chlorhexidine–cetrimide solution was the source of infection. Magalhães et al. [31] reported on a polyclonal bacteremia outbreak involving *B. cenocepacia* (formerly genomovar III) and *B. vietnamiensis* infection that occurred in a hemodialysis facility in Brazil. In our study, we found that *B. cepacia* was the predominant species in VAIs among hemodialysis patients. *B. cepacia* infection, although previously reported in cystic fibrosis patients, primarily spreads via cross-transmission [32]. Therefore, contamination with *B. cepacia*—possibly at the skin insertion site, catheter hub, or the contaminated intravenous fluid delivered through the catheter—may have caused the periodic transmission of *B. cepacia* in our institution.

BCC bacteria possess the ability to survive and proliferate in a water-based environment. BCC bacteria are historically described as major contaminants in both sterile and non-sterile pharmaceuticals, such as intravenous drugs, nasal sprays, and mouthwash, causing numerous nosocomial outbreaks registered during the past years [33]. Therefore, we speculate that the occurrence of the same strain of ST102 *B. contaminans* in our institution between May 22 and July 30, 2019, may have been caused by contamination in several ways. It is possible that the cause was due to sink drain contamination, in which the pathogen was present in the water or in contaminated products placed in sinks. Alternatively, the use of contaminated chlorhexidine solution for skin disinfection before catheter insertion and during follow-up care may have caused the dissemination of *B. contaminans*. There is a high probability that contaminated pharmaceuticals, especially detergent, caused the BCC contamination. However, we did not find evidence of BCC contamination in an institutional survey. Nevertheless, easy cross-transmission with BCC, facilitating the rapid spreading of infections between wards, is alarming to us, and further precautions should be taken in the future. 

## 5. Conclusions

This six-year epidemiology surveillance study revealed that *B. contaminans*, *B. cenocepacia*, and *B. cepacia* were the predominant non-CF BCC infections, including VAIs and other clinical infections, at a single institution in Taiwan. The MLST analysis provided important insights into the diversity of the BCC population in our institution. Our study demonstrated the high genetic diversity of *B. cenocepacia* in different clinical infections, providing its ability to survive in the healthcare setting’s harsh environment and to cause infection in different kinds of diseases. In contrast, single clonal ST102 *B. contaminans* in different diseases suggested the successful evolution of this strain to survive a wide distribution in clinical environments. Overall, the first insight into BCC’s phylogenetic diversity in healthcare environments will contribute to a greater understanding of their ecological role and evolution and provides insight into the BCC species’ potential clinical impact in non-CF infections. 

## Figures and Tables

**Figure 1 microorganisms-09-00051-f001:**
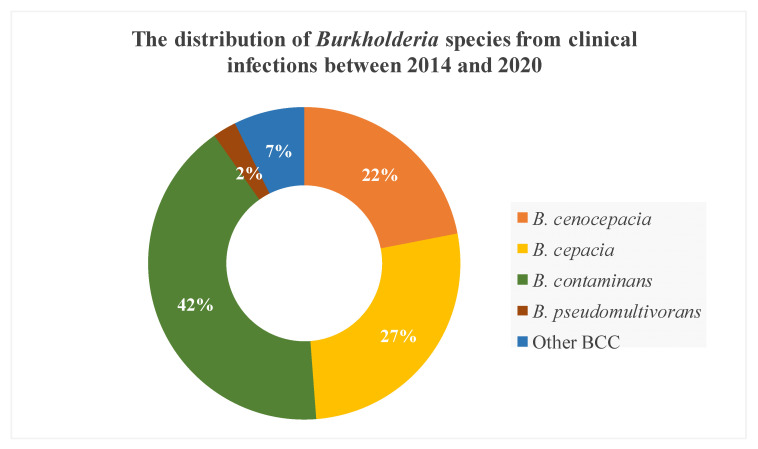
The species distribution of *Burkholderia cepacia* complex (BCC) from different clinical infections between 2014 and 2020.

**Figure 2 microorganisms-09-00051-f002:**
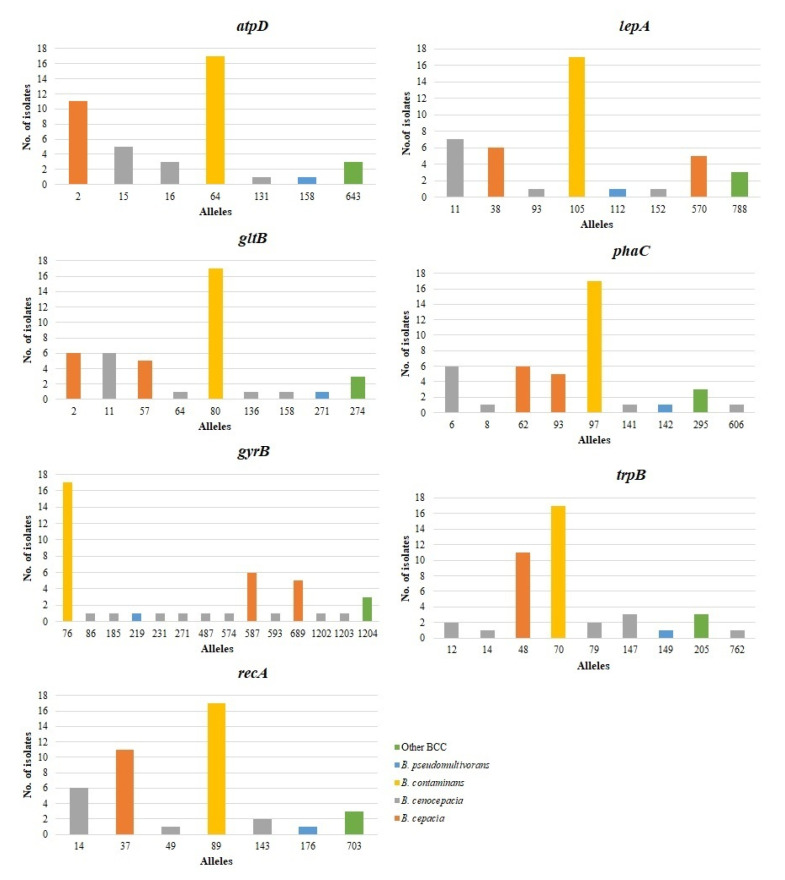
Frequency of alleles among the seven loci examined. For each locus, the number of times each allele occurred in different *Burkholderia cepacia* complex (BCC) species is shown.

**Figure 3 microorganisms-09-00051-f003:**
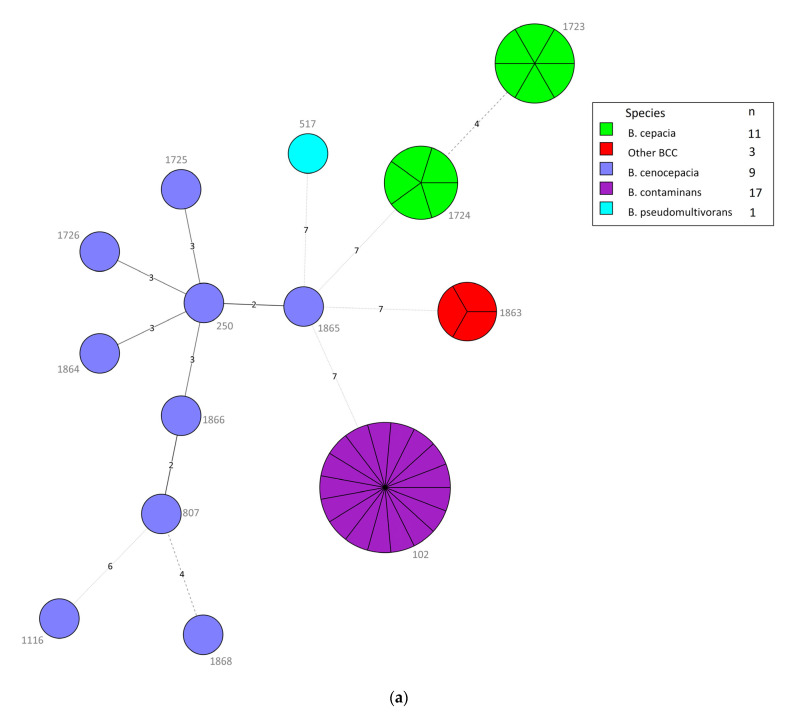
(**a**) Minimum spanning tree (MST) constructed with multilocus sequence typing (MLST), using BioNumerics version 7.6, of the 41 *Burkholderia cepacia* complex (BCC) isolates consisting of different species. Each node within the tree corresponds to a single sequence type (ST). The ST number is labeled next to the node. Each color inside the nodes represents the species of BCC. The size of the node is proportional to the number of isolates found with that profile. The length of branches between each node indicates the number of alleles (out of seven MLST genes) differing between two linked nodes/ST. (**b**) MST analysis of 41 BCC isolates depending on types of cases (i.e., vascular access infections (VAIs) and other clinical infections). Each color inside the nodes represents the types of cases.

**Table 1 microorganisms-09-00051-t001:** Clinical demographics of patients with *Burkholderia cepacia* complex (BCC) infections.

	VAI (*n* = 10)	Other (*n* = 26)
Sex of patients		
Male	4	17
Female	6	9
Age (median ± SD)	64.55 ± 17.48	72.65 ± 13.67

**Table 2 microorganisms-09-00051-t002:** The distribution of *Burkholderia* species in different clinical infections.

	VAI (*n* = 14)	Other (*n* = 27)
**Species**		
*B. contaminans* (*n* = 17)	3	14
*B. cepacia* (*n* = 11)	11	0
*B. cenocepacia* (*n* = 9)	0	9
*B. pseudomultivorans* (*n* = 1)	0	1
Other BCC (*n* = 3)	0	3
**Types of bacterial infection**		
Multispecies infection w/BCC	1	11
BCC only	13	16

**Table 3 microorganisms-09-00051-t003:** Nucleotide and allelic diversity of the seven loci in the 41 *Burkholderia cepacia* complex (BCC) isolates.

Locus	Length (bp)	G + C Content (%)	No. of Alleles	No. of Polymorphic Sites (S)	Average *d_N_/d_S_* Ratio	Nucleotide Diversity (π)	No. of Haplotypes (h)	Haplotype Diversity (Hd)	Total Number of Mutations (Eta)
*atpD*	443	62.1	7	23	5.5517	0.01227	7	0.748	27
*gltB*	400	67.4	9	32	4.2545	0.01821	9	0.782	34
*gyrB*	454	63.0	14	70	9.8871	0.04375	14	0.8	75
*lepA*	397	65.6	8	37	0.0478	0.02762	8	0.774	41
*phaC*	385	61.1	9	38	0.0252	0.02238	9	0.782	39
*recA*	393	67.5	7	52	0.01984	0.04532	7	0.774	58
*trpB*	301	69.4	9	31	7.52381	0.03161	9	0.757	37

**Table 4 microorganisms-09-00051-t004:** The distribution of sequence type (ST) of different species of *Burkholderia cepacia* complex.

Species	ST	Disease Type	Specimen	No.
*B. cenocepacia* *n* = 9	807	Other	Corneal ulcer	1
	1116		Urine	1
	1725		Blood	1
	1726		Sputum	1
	1864		Bronchoalveolar lavage	1
	1865		Wound	1
	1866		Sputum	1
	250 (CC31)		Tissue	1
	1868		Sputum	1
*B. cepacia* *n* = 11	1723	VAI	Blood	1
			Tip (Hickman)	5
	1724		Blood	2
			Tip (Hickman)	3
*B. contaminans* *n* = 17	102	Other	Blood	3
			Corneal ulcer	2
			CVP tip	1
			Endocervix discharge	4
			PCN	2
			Pus	2
		VAI	Blood	1
			Tip (Hickman)	2
*B. pseudomultivorans*	517	Other	Sputum	1
*n* = 1				
Other BCC	1863	Other	Pleural effusion	1
*n* = 3			Sputum	2

## Data Availability

All data generated or analyzed during this study are included in this published article.

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
