# Peer review of "Comparison of Microbiological Characteristics and Genetic Diversity between *Burkholderia cepacia* Complex Isolates from Vascular Access and Other Clinical Infections"

_microorganisms, 2020, doi:10.3390/microorganisms9010051_

Round 1
Reviewer 1 Report
Burkholderia cepacia complex (Bcc) bacteria are opportunistic pathogens best known for causing nosocomial infections. The goal of this study was to test samples from patients in Taiwan with vascular access infection and other non-cystic fibrosis patients to determine which Bcc species were responsible for these infections, as determined by comparing recA gene sequences and multilocus sequence typing (MLST). The study appears to have been performed using standard techniques, although the sample size was only 41 isolates. These finding would be of interest to the Burkholderia community, but it is unclear how these findings advance or confirm the current dogma regarding these infections. Thus, there are a few minor issues that need to be addressed before the findings can be completely assessed by the readership.
General issues
- Page 2, lines 53-54. There appears to be some words missing as the sentence does not make sense.
- Page 3, line 96. Please italicize the genus/species names.
- Page 4, lines 137-147. There are multiple genus/species and gene names that need to be italicized in this paragraph.
- Page 6, Figure 2 legend. Please italicize gene/species names.
- Discussion/Conclusion. It is not clear to this reviewer what are the major findings in this study that changes/advances/confirms what is already known about diseases caused by Bcc bacteria. What is significant about these findings?
- Bibliography. The entire section lacks italics for the bacterial genus/species in these references.
Author Response
Reviewer 1:
Point 1: Page 2, lines 53-54. There appears to be some words missing as the sentence does not make sense.
Response 1: We have modified the sentence to make it clear to understand.
[The species is epidemiologically linked to the contamination of medical devices and pharmaceutical products, such as nasal sprays, dialysis water, intravenous fentanyl, and moist washcloths [11-14], causing healthcare-associated infections.]
Point 2: Page 3, line 96. Please italicize the genus/species names.
Response 2: We have italicized the genus/species names.
Point 3: Page 4, lines 137-147. There are multiple genus/species and gene names that need to be italicized in this paragraph.
Response 3: We have italicized the genus/species names.
Point 4: Page 6, Figure 2 legend. Please italicize gene/species names.
Response 4: We have italicized the genus/species names.
Point 5: Discussion/Conclusion. It is not clear to this reviewer what are the major findings in this study that changes/advances/confirms what is already known about diseases caused by Bcc bacteria. What is significant about these findings?
Response 5: In this study, we demonstrated the prevalence of BCC species in different non-CF infections, including vascular access infections, and figured out this pathogen evolution in Asian populations. The non-CF BCC infections are newly clinical scenarios with few updated publications.
Point 6: Bibliography. The entire section lacks italics for the bacterial genus/species in these references.
Response 6: We have italicized the genus/species names.
Reviewer 2 Report
The present article entitled as "Comparison of microbiological characteristics and genetic diversity between Burkholderia cepacia complex isolates from vascular access and other clinical infections" reflects a 6-year single-institution study to (a) investigate the prevalence of BCC species from vascular access infections and other BCC infections (non-CF), (b) determine the genetic diversity and multilocus sequence of different bacterial species isolated from different infections and specimens, and (c) establish correlation between BCC species, sequence types (STs), and infection types.
Authors used in the article 41 BCC from 36 patients and I suggest to re-writte the conclusion section according to the abstract and the major outcomes obtained.
The outcome expressed in line 185-187 must be detailed in discussion sections with several hypothesis. The information (lines 241-246) is insufficient.
Lines 212-216 - this information must be changed for the results section?

Author Response
Point 1: Authors used in the article 41 BCC from 36 patients and I suggest to re-write the conclusion section according to the abstract and the major outcomes obtained.
Response 1: We re-wrote the conclusion section in this revised manuscript as the reviewer’s suggestion on p.9 (line 262-272).
[This 6-year epidemiology surveillance study revealed that B. contaminans, B. cenocepacia, and B. cepacia were the predominant non-CF BCC infections, including VAIs other clinical infections, at a single institution in Taiwan. The MLST analysis provided important insights into the diversity of the BCC population in our institution. Our study demonstrated the high genetic diversity of B. cenocepacia in different clinical infections, providing the ability to survive in healthcare settings’ harsh environment and cause infection in different kinds of diseases. In contrast, single clonal ST102 B. contaminans in different diseases suggested the successful evolution of this strain to survive a wide distribution in clinical environments. Overall, the first insight into BCC’s phylogenetic diversity from healthcare environments will contribute to a greater understanding of their ecological role and evolution and provides insight into the BCC species’ potential clinical impact in non-CF infections.]
Point 2: The outcome expressed in line 185-187 must be detailed in discussion sections with several hypothesis. The information (lines 241-246) is insufficient.
Response 2: We supposed more explanation in the discussion section on p. 9 (line 246-251).
[Therefore, we speculate that the occurrence of the same strain ST102 B. contaminans in our institution between May 22 and July 30, 2019, may have been caused by contamination in several ways. It is possible that cause by the sink drain contamination in which the pathogen’s presence in the water or contaminated products is placed in sinks—alternatively, the use of contaminated chlorhexidine solution for skin disinfection before catheter insertion and during follow-up care may cause the dissemination of B. contaminans. It is a high probability that contaminated pharmaceuticals, especially detergent, have caused the BCC contamination.]
Point 3: Lines 212-216 - this information must be changed for the results section?
Response 3: We provided more information in the result section so that the information is similar to the results section on p. 4 (line 147-150).
[In this study, 12 cases of multiple-species infections (i.e., isolation of BCC, especially B. contaminans and B. cenocapacia, and other species of bacteria in one specimen), was predominantly found in other BCC infections (11/27), especially from the endocervix discharge of B. contaminans infection.]
Round 2
Reviewer 1 Report
The authors did a good job of addressing my previous concerns. The only edits that need to be made are in lines 248-252, which is a rewritten section, as the English wording does not make complete sense